# Melatonin and Its Role in the Epithelial-to-Mesenchymal Transition (EMT) in Cancer

**DOI:** 10.3390/cancers16050956

**Published:** 2024-02-27

**Authors:** Carlos Martínez-Campa, Virginia Álvarez-García, Carolina Alonso-González, Alicia González, Samuel Cos

**Affiliations:** Department of Physiology and Pharmacology, School of Medicine, University of Cantabria and Instituto de Investigación Valdecilla (IDIVAL), 39011 Santander, Spain; alonsogc@unican.es (C.A.-G.); gonzalav@unican.es (A.G.); coss@unican.es (S.C.)

**Keywords:** melatonin, EMT, epithelial mesenchymal transition, carcinogenesis, cancer progression, signaling pathways

## Abstract

**Simple Summary:**

The epithelial-to-mesenchymal transition (EMT) is a process taking place during carcinogenesis. The phenotypic changes include the acquisition of new properties such as increased motility and polarity, leading to invasiveness and the formation of metastasis and chemo- and radioresistance. During the process, epithelial markers are lost whilst mesenchymal markers are overexpressed. EMT-related transcription factors are induced and multiple signaling pathways are activated. Several microRNAs are altered during the transition. Many of these molecules are regulated by melatonin, the pineal hormone, thus behaving as an inhibitor of the EMT in cancer progression. In this review, we summarize the present knowledge on the actions of melatonin on the EMT.

**Abstract:**

The epithelial-to-mesenchymal transition (EMT) is a cell-biological program that occurs during the progression of several physiological processes and that can also take place during pathological situations such as carcinogenesis. The EMT program consists of the sequential activation of a number of intracellular signaling pathways aimed at driving epithelial cells toward the acquisition of a series of intermediate phenotypic states arrayed along the epithelial–mesenchymal axis. These phenotypic features include changes in the motility, conformation, polarity and functionality of cancer cells, ultimately leading cells to stemness, increased invasiveness, chemo- and radioresistance and the formation of cancer metastasis. Amongst the different existing types of the EMT, type 3 is directly involved in carcinogenesis. A type 3 EMT occurs in neoplastic cells that have previously acquired genetic and epigenetic alterations, specifically affecting genes involved in promoting clonal outgrowth and invasion. Markers such as E-cadherin; N-cadherin; vimentin; and transcription factors (TFs) like Twist, Snail and ZEB are considered key molecules in the transition. The EMT process is also regulated by microRNA expression. Many miRNAs have been reported to repress EMT-TFs. Thus, Snail 1 is repressed by miR-29, miR-30a and miR-34a; miR-200b downregulates Slug; and ZEB1 and ZEB2 are repressed by miR-200 and miR-205, respectively. Occasionally, some microRNA target genes act downstream of the EMT master TFs; thus, Twist1 upregulates the levels of miR-10b. Melatonin is an endogenously produced hormone released mainly by the pineal gland. It is widely accepted that melatonin exerts oncostatic actions in a large variety of tumors, inhibiting the initiation, progression and invasion phases of tumorigenesis. The molecular mechanisms underlying these inhibitory actions are complex and involve a great number of processes. In this review, we will focus our attention on the ability of melatonin to regulate some key EMT-related markers, transcription factors and micro-RNAs, summarizing the multiple ways by which this hormone can regulate the EMT. Since melatonin has no known toxic side effects and is also known to help overcome drug resistance, it is a good candidate to be considered as an adjuvant drug to conventional cancer therapies.

## 1. Introduction: Melatonin, an Antineoplastic Molecule

For nearly 30 years, many cancer research groups have generated a huge amount of experimental data evidencing that melatonin, a hormone synthesized and released by the pineal gland following a circadian pattern, regulates cancer at different stages, ranging from the initial cell proliferation to invasion and metastasis. Interestingly, the pineal hormone interferes with a great number of biological processes such as sleep–wake rhythms, circadian timing, immune system regulation, oxidative stress, cytokine production or sexual maturation, with multiple intracellular signaling pathways being involved [1]. The first studies in melatonin and cancer were primarily focused on hormone-dependent tumors, mainly breast cancer [2], but it was later demonstrated that melatonin could also influence many other hormone-independent cancer types including pancreatic, gastric, colon, non-small-cell lung carcinoma or leukemia [3]. In the specific case of breast cancer, it has been well established that the main effects of melatonin are primarily directed toward inhibiting the estrogen-mediated signaling pathways at least through three different but somehow related mechanisms. Melatonin is known to downregulate the hypothalamic–pituitary–reproductive axis, leading to a decrease in the production of estrogens in what is known as an indirect inhibition mechanism. However, melatonin has also been reported to display a variety of effects on tumor cells. The pineal hormone can inhibit the enzymes involved in estrogen synthesis; therefore, melatonin is considered a selective estrogen enzyme modulator (SEEM). Additionally, melatonin can have a direct effect on tumor cells by impairing the transcriptional activity of the complex formed by estradiol and the estrogen receptor, thus behaving as a selective estrogen receptor modulator (SERM). Since many breast cancer treatment strategies include the use of compounds with either SERM or SEEM properties, these actions of melatonin suggest that this molecule has great potential in the prevention and treatment of estrogen-dependent tumors [4,5].

Many studies have demonstrated that concomitant administration of melatonin with some chemotherapeutic agents increases cancer cell sensitivity to these conventional drugs while limiting their adverse side effects on healthy cells. Most importantly, the pineal hormone has also been proven to reverse drug resistance in different cancer types [1,6]. Other antineoplastic actions carried out by melatonin include the inhibition of angiogenesis and other biological processes involved in tumor spread (metastasis) such as the epithelial-to-mesenchymal transition (EMT), a dynamic process consisting of the transition of a tumor cell from an epithelial state into a more mesenchymal phenotype, therefore resulting in changes in cell polarity, mobility and functionality. The EMT ultimately facilitates the release of tumor cells from the main tumor bulk, thus contributing to the progression from in situ carcinoma to invasive carcinoma [7].

In this revision, we will highlight the importance of the epithelial-to-mesenchymal transition in cancer malignancy and spread, emphasizing the role of the specific markers (adhesion molecules and key transcription factors) involved in this process to later summarize the current knowledge about the role of melatonin as an inhibitor of the EMT in different tumor types based on experimental, epidemiological and clinical studies. Lastly, future perspectives in the field will be discussed.

## 2. Stages of Carcinogenesis

The sequence of molecular events and biological processes leading to the development of cancer is known as carcinogenesis. The term “cancer” implies an accelerated and uncontrolled cell division that ends up compromising the homeostasis of the healthy surrounding tissues. Tumor cells show two distinctive hallmarks: the ability to escape the organism’s attempts to inhibit uncontrolled proliferation and the ability to infiltrate and invade healthy tissues. The three stages involved in carcinogenesis, as described in the three-step theory, imply that a healthy normal cell must undergo three sequential phases known as initiation, promotion and progression for the cells to become malignant. Interestingly, EMT plays a role in all three of them [8].

*Initiation*. The irreversible stage of initiation is triggered by DNA alterations occurring in a healthy cell. Once the switch from normal to cancerous cells has occurred, subsequent DNA mutations will help maintain the malignancy of the cell. Mutations usually activate oncogenes and inactivate tumor-suppressor genes [9,10]. Mutations can arise spontaneously or be originated by a variety of exogenous factors such as viral infections, chemical agents, ionizing radiation, tobacco or alcohol. Carcinogens alter the chemical structure of the nucleotides in the genome, and when these alterations escape DNA-repair mechanisms, the DNA sequence will be subsequently misread and passed along after the next division cycle [11].

The initiation phase can also be triggered by epigenetic changes. For example, DNA methylation can lead to the activation of proto-oncogenes and the inactivation of tumor-suppressor genes. Therefore, when proto-oncogenes like *c-Myc* or *Ras* are activated, abnormal proliferation leading to uncontrolled cell division occurs. On the contrary, when tumor-suppressor genes such as *Rb*, *PTEN* or *p53* are repressed, an increased mutation rate will occur, enabling cells to acquire a greater survival capacity. The importance of tumor-suppressor genes is highlighted by the fact that mutations in the tumor-suppressor *p53* (encoded by the *TP53* gene), causing its loss of function, are present in more than 50% of tumors of all types [12]. Alterations in the *TP53* sequence prevent cell cycle arrest, therefore promoting proliferation, and consequently, a large population of highly proliferative cells will be created.

*Promotion*. This stage of tumor development previous to tumor progression requires the involvement of certain molecules known as inductors or promoters that are usually activated once the transformation process is initiated. Promoters allow the descendants of transformed cells to survive, proliferate and escape apoptosis, thus leading to clonal growth [13]. However, promoters do not usually have mutagenic effects. Sex hormones like estrogens and androgens, other hormone types including glucagon and prolactin or certain chemical or physical agents such as UV light are examples of promoters [11]. The carcinogenic process implies that the cells that have been transformed by DNA alterations usually acquire survival advantages compared to the surrounding normal cells, including mainly higher proliferation rates and the evasion of apoptosis. Therefore, after the tumoral cell enters this phase, a large population of selected clonal cells displaying advantageous survival skills will be originated. As opposed to the initiation step, the promotion phase is usually reversible and the time that elapses between these two stages can be prolonged [12].

*Progression*. Further exposure to promoters and carcinogenic agents will cause the cells to acquire driver mutations. The progressive accumulation of DNA alterations makes the tumor cells more malignant, invasive and resistant to treatments [14]. These driver mutations allow transformed cells to gain growth advantages over time [15].

More than a hundred genes have been identified as key players in the process of cancer progression. These relevant genes can be grouped into several categories according to their function. Some of these genes encode for the proteins involved in cell-survival processes, and this group includes growth factor receptors (*EGFR* and *FGFR*) and the proteins involved in the Ras signaling pathway (*KRAS* and *BRAF*) and the PI3K signaling pathway (*AKT*, *mTOR* and *PTEN*). Some other genes are classified as genome-maintenance genes (*p53* and *BRCA1*). A third category includes genes participating in adhesion and proteolytic processes (*CTNNA2* and *ADAMTS18*), actin cytoskeleton remodeling (*RAC1* and *ROCK1*) and genes that regulate the EMT (*APC1*, *TWIST1*) through different mechanisms [16]. Moreover, the tumor microenvironment also contributes to stimulating cancer migration and invasion by secreting different protumorigenic proteases, cytokines and growth factors in response to hypoxia, nutrient depletion, chemoattractants and extracellular matrix rigidity [17,18]. As the tumor bulk grows, some tumor microenvironment components including fibroblasts, macrophages and immune-suppressing cells infiltrate the tumor in order to modulate immune responses. During the progression phase, both the extracellular matrix composition and structure become altered, and this process, coupled with the cellular necrosis due to macrophage activity, will lead to tissue damage, dysfunction and ultimately cell death [19].

An important cytokine secreted during the invasion process is the vascular endothelial growth factor (VEGF), which is considered the main factor triggering angiogenesis; that is, the formation of new blood vessels. VEGF stimulates the proliferation and differentiation of endothelial cells through binding to its receptor VEGFR [20]. New blood vessel formation provides a nourishing environment to the growing tumor by increasing the delivery of oxygen and nutrients and the removal of waste products to meet the increasing demands of thriving malignant cells. Interestingly, tumor angiogenesis also provides the migration tracks often used by malignant cells to escape away from their original location, infiltrate further tissues and form metastasis [21]. Another mechanism used by tumor cells to increase their malignancy and enhance their invasion properties is related to their cellular plasticity and the ability to dedifferentiate; that is, losing their specialized properties in order to adopt a stem-like phenotype [22]. This stem-like feature can be achieved following the activation of the EMT program, a phenomenon of great importance for the acquisition of migration and invasion properties often associated with an increase in motility coupled with a loss of cell-to-cell junctions, enabling malignant cells to complete most of the steps of the invasion–metastasis cascade. Thus, several biological and molecular mechanisms seem to be involved in the acquisition of malignant features such as increased cell motility and invasion properties, cell death resistance, chemo- and radiotherapy resistance and tumor recurrence [23]. The activation of the EMT is triggered by a number of key transcription factors, each aimed at activating a variety of intracellular signaling pathways, as reviewed in the next section [24].

## 3. The Epithelial-to-Mesenchymal Transition

The epithelial-to-mesenchymal transition is a biological process experienced by epithelial cells that encompasses several morphological and functional changes that ultimately result in the acquisition of distinctive cell characteristics including the loss of cell adhesions and the increase in migratory and invasive properties. The activation of an EMT program can drive epithelial cells into a more mesenchymal state, allowing cells to acquire many of the traits associated with mesenchymal cells, including an increase in motility and changes in cell polarity, and they become more resistant to cell death and are more prone to disseminate [25].

Even if it is intrinsically related to cancer, the EMT phenomenon occurs in some other physiological events such as developmental processes related to embryogenesis, organ fibrosis and wound healing. In order for the EMT to take place, cells must be transcriptionally reprogrammed, and that process is carried out by a panel of key transcription factors. According to the biological context in which they develop, EMT programs can be classified into three different subtypes [26].

A type 1 EMT occurs during embryo implantation, and it is crucial for subsequent organ development, playing a pivotal role in gastrulation and neural crest formation [27]. The possibility of the cell reversing the process in terms of mesenchymal cells transforming back into the epithelial type through a mesenchymal-to-epithelial transition (MET) is likely for type 1 MET-undergoing cells. As a result of reversing the EMT program, a secondary epithelium can be generated during embryogenic development [26,28].

A type 2 EMT is associated with wound healing, tissue repair processes, tissue regeneration and organ fibrosis. During wound healing, this type of EMT induces changes in cell physiology in order to gain motility and allow for organ regeneration in response to damage. A type 2 EMT is a process tightly associated with inflammation, and it only concludes when the inflammatory process attenuates. During organ fibrosis, type 2 EMTs can continually respond to inflammation, and that can lead to organ destruction. Several EMT biomarkers have been described to orchestrate this process, including TGF-β [26,29,30].

Type 3 is the most extensively studied EMT type since it is the one involved in carcinogenesis. It occurs in neoplastic cells that have previously undergone a series of genetic and epigenetic alterations in specific genes involved in cell proliferation that will lead to the acquisition of a number of phenotypic traits associated with an increase in cell proliferation, migration and invasion. Unlike the other EMT types, cells undergoing a type 3 EMT program acquire the ability to escape the control mechanisms present in the organism, thus favoring malignancy and tumor progression. Interestingly, tumor cells can undergo this process partially, with some cells retaining several epithelial characteristics, while others will fully commit to the acquisition of the mesenchymal phenotype [26]. In this review, we will focus our attention on type 3 EMTs and how melatonin can influence this process.

The last step in the invasion–metastasis cascade inevitably requires a certain degree of cell plasticity, and it eventually involves the activation of MET programs, allowing the cells to transition between different epithelial and mesenchymal states along the spectrum [26,28,31]. Since type 3 EMTs play a key role in tumor development and spread, in the next section, we will discuss the most relevant features of this particular EMT type, exploring the different intracellular signals and molecular mechanisms engaged in this process [32].

## 4. Molecular Mechanisms of Epithelial-to-Mesenchymal Transition in Cancer Progression

The EMT was initially described as an “epithelial to mesenchymal transformation”, but soon the process was renamed as the epithelial-to-mesenchymal transition to remark its transient nature. This phenomenon was initially perceived as a shift between two possible states, epithelial or mesenchymal, but the change from “transformation” to “transition” reflected the existence of an array of transitional states and cell plasticity, so nowadays cancer cells are no longer considered to switch from “totally epithelial” to “totally mesenchymal”; instead, they are thought to transition through a range of intermediate states [24,33]. The ability of epithelial tumor cells to partially or fully acquire mesenchymal characteristics and eventually regain their original features reveals the inherent plasticity of the epithelial cells. The most significant morphological changes that these cells undergo during the EMT include the loss of the apical–basal polarity, loss of cell-to-cell junctions, reorganization of the cytoskeleton and changes in cell shape. All these newly gained features are obtained by the extensive reprogramming of gene expression that ultimately leads to increased motility and the acquisition of an invasive phenotype [24,32,33,34]. The markers, transcription factors and signaling pathways participating in the transition are summarized in Figure 1.

The EMT is regulated at different levels, including transcriptional modifications, epigenetic changes and alternative splicings of certain EMT effectors. The transition of a cell from epithelial origin to a mesenchymal one involves the loss of a series of markers and the gain of others during the transition toward a more mesenchymal phenotype. Perhaps the most extensively described molecular changes occurring during the EMT are related to the functional loss of the proteins involved in cell–cell adhesions such as the adhesion molecule E-cadherin, the *zonula occludens* protein (ZO-1), cytokeratin and the extracellular matrix proteins laminin-1 and type IV α1-collagen [24,34,35].

Type 3 EMTs are primarily initiated by the coordinated activation of a small group of EMT master regulators including Snail1, Snail2 (Slug), Twist1, ZEB1 and ZEB2. The activity of these zinc-finger transcription factors results in cellular adhesion disassembly, the loss of polarity and an acquired motile mesenchymal phenotype [35,36]. The regulation of the genes involved in the EMT process is achieved by the cross-regulation of these transcription factors, and they often functionally cooperate to target specific gene promoters. EMT transcription factors also drive the repression of the genes involved in the maintenance of epithelial features while simultaneously activating genes conferring the cells’ mesenchymal characteristics. Therefore, the same transcription factors are involved in both the activation and repression processes [37,38]. The Snail superfamily of zinc-finger transcription factors includes Snail1, Snail2 (Slug) and Snail3 (Smug), the first two being involved in both physiological and cancer-associated EMTs. The overexpression of Snail is often associated with chemoresistance, and the underlying mechanism driving this process involves the acquisition of a multidrug resistance phenotype (MDR) through activation of the drug efflux transport P-glycoprotein (P-gp) [39]. The zinc-finger E-box binding proteins 1 and 2 (ZEB1 and ZEB2) have been associated with malignancy in several types of cancer. ZEB1 is reportedly a key regulator of the EMT [40]; an increased transcription of ZEB1 induces cell plasticity and plays a role in the transition from a nontumorigenic state to a tumorigenic state. ZEB1 actions are directly linked to the functional loss of E-cadherin, a hallmark of the EMT, by the inhibition of the transcriptional expression of the E-cadherin gene [40]. ZEB1/ZEB2 coexpression is an indicator of poor prognosis in head and neck cancer [41]. ZEB2 is expressed in a wide variety of tumors including ovarian, breast, pancreatic and gastric cancers, among many others. Besides its role in regulating the EMT, the overexpression of ZEB2 in cancer cells results in the deregulation of the cell cycle, inhibition of apoptosis, increased cell proliferation and cancer progression [42]. Twist1 is a transcription factor bearing a basic helix–loop–helix domain that plays a crucial role during embryogenesis and in mesoderm formation and differentiation [43]. In cancer, Twist1 promotes angiogenesis, and it has been reported that Twist1 overexpression correlates with an increased expression of the vascular endothelial growth factor (VEGF) and angiopoietin-2 (Ang-2) genes in breast cancer cells [44]. The overexpression of Twist1 is also associated with the upregulation of the gene encoding for the multidrug resistance protein (P-gp) expression. Furthermore, it has been reported that the knockdown of Twist1 sensitizes cancer cells to chemotherapy [45]. Regarding the EMT, it has been well established that Twist1 is one of the master regulators of this process. Twist1 is directly involved in the upregulation of N-cadherin gene expression by directly binding to its promoter, hence resulting in its transcriptional activation [46,47].

Upon the activation of the EMT program, malignant cells acquiring a more mesenchymal phenotype undergo changes in their gene expression pattern, mainly increasing the expression of markers such as N-cadherin; O-Cadherin; different types of integrins (α5β1 and αVβ6); vimentin; FSP1; β-catenin; certain extracellular molecules such as α-1 I and III collagen, fibronectin and laminin-5 [48].

The EMT is also subjected to microRNA regulation (Figure 2). Many miRNAs are known to have a role in the repression of the EMT process by regulating the expression of certain transcription factors. Thus, Snail 1 is repressed by miR-29, miR-30a and miR-34a [49,50,51]. miR-1 and miR-200b have been shown to interact with Slug in a self-reinforcing regulatory loop so that the induced expression of miR-1 or miR-200b leads to the depletion of Slug, thus inhibiting the EMT [52]. Additionally, the expression of ZEB1 and ZEB2 is repressed by miR-200 and miR-205. In cells undergoing EMT after the activation of the TGF-beta signaling pathway, miR-200 and miR-205 appear to be downregulated, and consequently, both ZEB1 and ZEB2 are upregulated [53]. As opposed to these examples, some other miRNAs are known to promote the EMT. Thus, miR-21 has been reported to act as an oncogenic microRNA given its protumorigenic effects [54]. Exosomal miR-21 derived from multiple myeloma cells has been reported to play a role in EMT through the activation of the TGF-β/SMAD7 signaling pathway [55]. In breast cancer, an increase in miR-9 expression leads to the repression of CDH1 (E-cadherin), therefore increasing cell migration and invasion [56]. The accelerated maturation of miR-10a by XRN2 also promotes tumor migration and invasion by stimulating the EMT in a lung cancer experimental model [57]. miR-10a is another example of a miRNA involved in the activation of the EMT by stimulating the Hippo signaling pathway in a mouse xenograft tumor model implanted with pancreatic cells [58]. miR-135b seems to play a role in controlling the EMT in ovarian cancer by regulating p-Akt and PTEN expression levels, resulting in the alteration of EMT-associated proteins such as E-cadherin, N-cadherin, Snail and vimentin [59]. In ovarian cancer cells, miR-150-5p upregulation also affects the EMT by activating the miR-150-5p/c-Myb/Slug axis, and the inhibition or this miRNA significantly inhibits biological processes closely related to the EMT such as cell migration and invasion [60].

Lastly, it is worth mentioning that some microRNAs that appear to be involved in the regulation of the EMT exert their functions downstream of the EMT master TFs. That includes miR-155, a repressor controlling the expression of RHOA and resulting in the loss of tight junctions. miR-155 is also directly regulated by TGFβ-mediated signaling [61]. In metastatic breast cancer cells, the EMT master TF Twist1 binds the promoter of miR-10b, which has been classified as a metastamir, increasing miR-10b gene expression and consequently stimulating both cell migration and invasion [62]. TGFβ also induces the upregulation of miR-24 expression, which has been reported to play a role in the regulation of neuroepithelial cell-transforming Net1A, an activator of RhoA. These findings highlight the importance of miR-24 in the regulation of the EMT program as it seems to be involved in the disruption of *adherens* and tight junctions [63]. Interestingly, Twist1 upregulates miR-10b expression, resulting in the loss of E-cadherin, while on the other hand, a decrease in Twist1 expression leads to the reduction in miR-10b expression, causing E-cadherin levels to be restored and therefore reducing invasiveness [64].

## 5. Signaling Pathways Involved in the Epithelial-to-Mesenchymal Transition

Current knowledge about the EMT has revealed that the transition from an epithelial to a mesenchymal phenotype is a complex process involving multiple and dynamic states in which many signaling pathways participate. Therefore, the activation of the EMT program will occur only after some stimulating signals are triggered [24].

*TGF-β signaling*. Transforming growth factor-β (TGF-β) is a ubiquitously expressed cytokine that acts as a key regulator of embryonic development and postnatal homeostasis. TGF-β is secreted as a latent factor that is sequestered by proteins of the extracellular matrix, and after its release and subsequent activation, it binds to cell-surface receptors in target cells [65]. Interestingly, TGF-β exerts different actions depending on the stage of tumor development. At early stages, TGF-β represses tumor growth, but when the cancer progresses, TGF-β exerts an opposite action by promoting tumor proliferation and metastasis. This pro-oncogenic effect is achieved by the activation of several intracellular processes, with the initiation of the EMT being one of the key ones [66]. TGF-β can activate a variety of signaling cascades after binding its specific TGF-β receptors 1 and 2 (TGFβR1 and TGFβR2). The activation of any of the receptors will trigger the phosphorylation of Smad2/3, which will eventually be involved in the induction of the EMT program after binding to Smad4. However, transcriptional regulation mediated by TGF-β can also be achieved by alternative mechanisms within the Smad2/3 pathway. Therefore, intracellular signaling triggered by TGF-β can be classified into Smad-dependent or Smad-independent TGF-β pathways [67]. The most representative transcription factors regulating the EMT, such as Snail, Slug, ZEB1, ZEB2 and Twist1, are considered direct Smad2/3 targets. Other relevant EMT-related transcription factors including Paired-related Homeobox 1 (PRX1), Forkhead box C2 (FOXC2), Forkhead box A2 (FOXA1) and High Mobility Group AT-hook 2 (HMGA2) also seem to be regulated by Smad2/3 [68]. Remarkably, the downregulation of Smad4 expression does not initiate cancer, but it promotes tumor progression initiated by other genes such as KRAS in pancreatic and colorectal cancer [67,69].

*Wnt signaling*. The protein Wingless (Wnt) is considered to be a key player in the induction of the EMT. The Wnt signaling pathway is widely known to regulate crucial processes in the cell including cell proliferation, cell-to-cell adhesion, polarity and motility, and therefore, this signal transduction cascade coordinates embryo development, stem cell renewal and adult tissue homeostasis. Upon binding its membrane receptor, Wnt-mediated signaling can be transduced into two different intracellular signaling pathways: the canonical Wnt/β-catenin pathway, in which β-catenin is involved, and a noncanonical pathway, which is β-catenin-independent. The β-catenin protein is the central molecule and the key nuclear effector of the canonical Wnt pathway. β-catenin is not only a transcription factor, but it also functions as an important component of the cytoskeleton. In the absence of Wnt ligands, ubiquitin-dependent proteasome degradation keeps the β-catenin protein at its basal levels [70]. Within the canonical pathway, Wnt is a ligand for a specific receptor complex formed by the Frizzled (Fz) protein and a low-density lipoprotein-related protein (LRP5/6). The binding of Wnt to its receptor complex induces the phosphorylation of Dishevelled (Dvl), which in turn inhibits the complex formed by Glycogen Synthase Kinase 3 beta (GSK-3β), Axin, Adenomatous Polyposis Coli (APC) and Casein Kinase 1 (CK1). This complex is responsible for the ubiquitylation and degradation of β-catenin by the proteasome. Therefore, the inhibition of this complex prevents the degradation of β-catenin, and in turn, its cytoplasmic levels will be increased. The accumulation of stabilized β-catenin in the cytoplasm leads to its translocation into the nucleus, where it acts as a transcriptional regulator [71]. The overactivation of the Wnt/β-catenin signaling pathway triggers the epithelial–mesenchymal transition by stimulating the expression of EMT-related core transcription factors. For example, in colorectal cancer (CRC), enhanced Wnt/β-catenin signaling increases the levels of SNAIL, which in turn represses E-cadherin transcription and consequently promotes local invasion [72]. It has been demonstrated that β-catenin directly interacts with the epithelial marker E-cadherin, creating strong cell–cell junctions. Decreased E-cadherin levels impact the Wnt/β-catenin signaling by sequestering β-catenin in the cytoplasm [73]. This signal triggers the activation of EMT once β-catenin enters the nucleus. Simultaneously, and secondary to an EMT-derived effect, E-cadherin expression is reduced together with an increase in fibronectin production and Slug and Twist upregulation. Consequently, the loss of *adherens juctions* not only modifies the cytoskeletal composition, but it also allows tumor cells, now displaying more mesenchymal features, to invade the extracellular matrix and migrate to surrounding parenchyma along the newly formed matrix of fibronectin [74].

*Notch signaling*. The Notch-signaling transduction pathway is a highly conserved mechanism of signal transduction in most animals. Notch proteins are a family of transmembrane proteins comprising four notch receptors in mammals, referred to as Notch1 to Notch4. These proteins act as membrane receptors capable of transducing extracellular signals into a cascade of intracellular events that will eventually lead to the regulation of gene expression, controlling key cellular processes. Notch receptors are heterodimers formed by a large extracellular domain and a shorter intracellular component. Humans and mice have five acknowledged NOTCH ligands, and all of them display unique and redundant functions in the cell. When the ligands Jag-1 or Jag-2 bind to the receptor, the intracellular domain is cleaved by γ-secretase [75]. The activated Notch receptor translocates to the nucleus and acts as a transcription factor, promoting the expression of the nuclear factor-kappa B (NF-κB) gene, which has been shown to play a key role in tumor growth and progression. Notch signaling has also been shown to induce the activation of the Akt signaling pathway [74]. Additionally, the activation of Notch signaling has been reported to influence Snail and Slug expression, both considered master EMT TFs. Interestingly, Slug is essential for the Notch-dependent repression of E-cadherin expression as well as β-catenin activation [76].

*IL6/STAT3 signaling*. The cytokine interleukin-6 (IL-6) is very often present at the tumor–normal tissue interface as it is secreted by cancer cells but also by other cellular components of the tumor microenvironment such as endothelial cells or tumor-infiltrating macrophages. One of the effects of the release of IL-6 in the tumor is the activation of the STAT3 signaling pathway in malignant cells, altering cell-to-cell adhesions and promoting a number of cellular processes such as increased cell proliferation, enhanced motility and the activation of the EMT program [77]. The ability of IL-6 to trigger the EMT has been observed in many types of cancers including breast, liver, colon, head and neck and laryngeal squamous carcinoma [78]. There is a great amount of evidence about the role of the IL6/STAT3 signaling pathway influencing the expression of many EMT transcription factors, including Snail, Slug and Twist [78].

*PI3/AKT signaling*. The PI3K/AKT axis plays a crucial role in the initiation of the EMT process. When this pathway is activated by the phosphorylation of Akt, in response to various growth factors such as vascular endothelial growth factor (VEGF) or epidermal growth factor (EGF), it can induce metastasis and chemoresistance in several types of cancer. It is known that the phosphorylation of Akt leads to Snail activation and E-cadherin downregulation, thus contributing to the initiation of the EMT process and therefore promoting metastasis and treatment resistance in many different cancer types [79]. These malignant traits are reportedly carried out through the phosphorylation of the transcription factor Twist, which has been associated with E-cadherin repression, the increase in pro-oncogenic genes and a worsening cancer prognosis [80]. Interestingly, the phosphorylation of Akt/protein kinase B (PKB) mediated by Twist1 promotes EMT and metastasis in breast cancer cells by modulating the Twist transcriptional target TGF-β2. The expression of TGF-β2 leads to enhanced TGF-β receptor-mediated signaling, which in turn maintains a hyperactive phosphoinositide 3-kinase (PI3K)/Akt signaling [81].

*The Hedgehog-GLI signaling*. The Hedgehog (HH-GLI) signaling pathway is mainly related to embryo development, meaning that an alteration in any of the proteins participating in this pathway could lead to potential fetus malformation or abnormalities. The Hh pathway signaling occurs after the binding of any of the HH-GLI ligands to the transmembrane receptor Patched 1 (PTCH1), thus allowing for the activation of Smoothened (SMO), another transmembrane receptor coupled to G-proteins. The activation of the SMO receptor triggers a cascade of intracellular events that results in the activation of three glioma-associated oncogene (GLI) transcription factors (GLI1, GLI2 and GLI3) [82]. The HH-GLI signaling pathway can promote the EMT by upregulating the expression of master EMT transcription factors such as Snail, Slug, Twist2, ZEB1 and ZEB2 [83]. Additionally, through crosstalk with other signaling pathways (such as the TGF-β pathway), the HH-GLI intracellular signaling can induce the transcription of PTCH1 in a self-regulatory loop [84]. Moreover, the HH-GLI pathway has been associated with cancer stem cell properties, and the crosstalk between the NFκ-B and GLI1 pathways highlights the importance of GLI1 in promoting cell dedifferentiation and stemness derived from the activation of the PI3K/Akt/NFK-β signaling pathway [85].

*Receptor tyrosine kinase signaling*. The receptor tyrosine kinases (RTKs) belong to a family of membrane receptors involved in mediating a wide variety of complex biological functions, including the activation of the EMT in cancer when RTK signaling is dysregulated. Some of the most representative members of this family are the Epidermal Growth Factor Receptor (EGFR), the Insulin-Like Growth Factor Receptor (IGF-R), the Platelet-Derived Growth Factor Receptor (PDGFR), the Fibroblast Growth Factor Receptor (FGFR) or the Vascular Endothelial Growth Factor Receptor (VEGFR) [86]. The RTK receptor gets activated upon ligand binding, which induces the dimerization of the receptor. For most RTKs, activation results in the autophosphorylation of key Tyrosine residues, which in turn enables the recruitment of downstream signaling effectors. Growth factors that bind to RTKs trigger a variety of intracellular signaling pathways such as the PI3K/AKT, MAPK, ERK or JNK pathways, all of them involved in the EMT. Therefore, RTK-specific ligands have the potential to induce the EMT at least partially, driving a shift in cytoskeletal dynamics, promoting the loss of cell–cell adhesions and switching epithelial cells to a more mesenchymal phenotype [87]. For example, Snail1 gene expression is induced by FGFR and IGF1R-mediated signaling [88,89], IGFR can also increase ZEB1 protein levels [89] and EGFR decreases E-cadherin levels through the endocytosis of this molecule while simultaneously increasing Snail and Twist expression genes [90]. Additionally, VEGF, the main factor involved in the development of new blood vessels, triggers the EMT by binding the VEGF Flk1/KDR receptor (VEGFR-2) [20]. Experimental studies using tumor biopsies from gastric cancer patients have shown a correlation between VEGF and the expression of key EMT markers. More specifically, VEGF and E-cadherin expression levels seem to be inversely correlated while VEGF and N-Cadherin levels appear to be positively associated. These results highlight the contribution of VEGF to tumor growth and metastasis by promoting the EMT in gastric cancer [91].

## 6. Melatonin as an Inhibitor of the Epithelial-to-Mesenchymal Transition

Melatonin is the main product of the pineal gland. This synthesis of this indoleamine is mainly controlled by the hypothalamic suprachiasmatic nucleus following a circadian pattern, meaning that the release of this hormone typically peaks at night while reaching lower plasma levels during the daytime. Melatonin is an extremely lipophilic compound that can easily cross circulatory barriers. The pineal hormone is also considered a pleiotropic substance that performs an array of physiological functions: it plays a role in seasonal organisms; synchronizes circadian rhythms; promotes sleep; is considered an effective antioxidant compound; also displays neuroprotective, cardiovascular and immunomodulatory effects; is known to regulate hormone release; and it reportedly participates in body and bone mass regulation [92]. It is widely acknowledged that the pineal hormone can inhibit different types of cancer both in vitro and in vivo [1,2,3,4,5,93]. The relevant role of this indoleamine as an inhibitor of cancer growth was first documented in breast cancer, but it has also been studied in other hormone-dependent cancer types including ovarian and prostate tumors [2].

Experiments carried out in rodents have shown that melatonin reduces the initiation and growth of breast cancer whilst pinealectomy, which means the surgical removal of the pineal gland, has the opposite effect, highlighting the contribution of this hormone to the development of this particular type of tumor [4]. The oncostatic effect of melatonin on hormone-dependent tumors could be explained, at least in part, by the ability of the pineal hormone to inhibit the estrogen-dependent and androgen-dependent signaling, thus acting as a selective estrogen receptor modulator (SERM). Moreover, since melatonin controls the expression and activity of several enzymes participating in estrogen synthesis and degradation, this hormone is known to act as a selective estrogen enzyme modulator (SEEM) [2,4,94].

*Oncostatic actions of melatonin*. More recently, new evidence has emerged regarding the oncostatic role of melatonin in a broader spectrum of tumors. Among the oncostatic actions of this indolamine, it is worth mentioning its ability to slow down or even completely inactivate most of the biological processes that promote tumor development, such as tumor cell growth, cell invasiveness, angiogenesis, apoptosis and metastasis. Notably, these antitumor actions are achieved by a surprisingly broad variety of mechanisms. Some of these mechanisms are relative to the radical scavenging properties displayed by the pineal hormone [95,96] and the prevention of DNA damage either at the early stages of cancer or even later in the disease after cancer treatment with chemotherapy or radiotherapy [97,98]. Melatonin can also exert metabolic actions on cancer cells. As an example, in breast cancer models, melatonin has been proven to reduce tumor growth by promoting fatty acid uptake by the tumor cells [99]. Moreover, this hormone can reverse the increased rate of glucose uptake and glycolysis in cancer cells, as well as their preferential production of lactate (the Warburg effect), which is broadly displayed by many types of tumor cells [100]. The pineal hormone is also known to regulate cell fate decisions. Some studies have shown that melatonin can play a role in regulating cell cycles in cancer cells, mainly by inhibiting cell proliferation and promoting apoptosis [101,102]. More recently, the proapoptotic effect attributed to the pineal hormone has been linked to melatonin-mediated mitochondrial reverse electron transport, thus inducing reactive oxygen species (ROS) production [103]. Also, as reviewed by Markus et al. [104], the antitumor actions of the pineal hormone have been associated with the activation of the immune system by increasing the proliferation of some types of immune cells such as neutrophils and monocytes as well as by directly stimulating their phagocytic capacity. Interestingly, the disruption of circadian rhythms after the surgical removal of the pineal gland reduces the amount and activation of immune cells [105]. Melatonin has also been reported to suppress telomerase activity in breast cancer cells, thus contributing to cell death by preventing their genomic stability [106]. Lastly, the pineal hormone has been shown to play a role in angiogenesis, impairing the development of new vessels that can provide oxygen and nutrients to the tumor bulk [107].

Given all the different ways in which melatonin can contribute to reducing tumor growth and the fact that the pineal hormone is an endogenous compound with no potential side effects, melatonin seems to be a good candidate to be considered as an adjuvant to conventional cancer treatments. Notably, melatonin has been described to exert protective actions against the undesirable side effects derived from the use of chemotherapeutic agents not only by reducing their toxicity but also by increasing the efficacy of these drugs, therefore improving survival expectations and the quality of life of these patients [108]. Some in vitro studies have shown that melatonin treatments sensitize cancer cells to conventional drugs by a number of different mechanisms such as increasing apoptosis and inducing cell cycle arrest [6,109,110,111]. Furthermore, some lines of evidence indicate that the concomitant administration of melatonin in cancer patients increases the one-year survival rate and reduces radiotherapy-related toxicities [112]. Overall, these studies suggest that melatonin has great potential to be considered as an adjuvant to conventional cancer treatments.

*Effect of melatonin on the EMT-activating transcription factors*. Research conducted both in vitro and in vivo has demonstrated that melatonin acts at all different stages of cancer, from initiation to tumor development and metastasis. Many of the molecular mechanisms associated with the migration of tumor cells into the vascular system are inhibited by the pineal hormone [1]. In the past few years, it has been demonstrated that melatonin can inhibit different processes leading to metastasis, including the remodeling of the extracellular matrix, angiogenesis, migration and the epithelial-to-mesenchymal transition [113]. In this regard, there is a growing amount of evidence related to the actions of melatonin on different effectors and the signaling pathways involved in cancer-related EMTs. Snail, Slug, ZEB1, ZEB2 and TWIST1 are considered core EMT-TFs, and they are largely responsible for initiating the EMT. However, there is an array of different transcription factors such as PRX1, FOXC2, FOXA1, HMGA2, SOX9, SIX1 and YAP1 [67] that have been characterized as inductors of some EMT properties, thus cooperating with the core EMT-TFs [114]. As a result of the combined activity of all these transcription factors, EMT-undergoing cells experience an increase in the expression of important cell-adhesion proteins such as N-cadherin; O-cadherin; integrin; vimentin; β-catenin; and some extracellular molecules involved in cell migration processes including α-1 type I and type III collagen, fibronectin and laminin-5 [47]. Therefore, the coordinated action of EMT-related transcription factors activates the classic EMT program resulting in the disorganization of cellular adhesions, loss of epithelial polarity and manifestation of an acquired mesenchymal motile phenotype, usually leading to stemness and survival [35,36,114].

Research conducted by using different models of cancer has revealed the effect of melatonin on the key EMT regulators Snail and Slug. One of these studies, conducted in a model of gastric cancer, pointed at the pineal hormone as a regulator of Snail and Slug expression in parallel to the inhibition of the CCAAT/enhancer-binding protein *β* (C/EBP*β*). This investigation concluded that the pineal hormone may restrict gastric tumor growth and peritoneal dissemination in vivo [115]. Also, in human gastrocarcinoma, melatonin has been reported to suppress interleukin (IL)-1β-induced EMTs by increasing the expression levels of β-catenin and E-cadherin while reducing the levels of vimentin, fibronectin, Snail, MMP2 and MMP9 [116]. Additionally, in UC3 bladder cancer cells, a cotreatment with melatonin and valproic acid increased cytotoxicity by stimulating death-related signaling pathways, thus leading to synergistic growth inhibition while enhancing the expression of E-cadherin and decreasing Snail and Slug mRNA levels [117]. The inhibition of Snail and Slug and the upregulation of E-cadherin levels by melatonin has also been reported in cancer stem cells isolated from SKOV3 ovarian cancer cells, thus inhibiting EMT-associated features such as cell migration and metastasis [118]. A study conducted in an epithelial ovarian cancer (EOC) model showed that the pineal hormone significantly reduced the abdominal tumor burden of ovarian cancer induced by chronic restraint stress, a result explained, at least in part, by the inhibition of the NE/AKT/β-catenin/SLUG axis [119]. Similarly, the results derived from an in vivo study on oral cancer showed that the pineal hormone impaired cell proliferation and apoptosis resistance while upregulating Bax and downregulating Bcl-2. These effects led to a reduction in Snail-expression levels, which subsequently caused an increase in E-cadherin levels, thus impairing cell survival, motility, angiogenesis and invasion [120]. It has been reported that both in hormone-dependent and hormone-independent breast cancer cell models, melatonin and the chemotherapeutic agent taxol cooperate to inhibit the generation of ROS, induce antiproliferative effects and decrease invasion and metastasis. When the molecular bases of this cooperative action were explored, it was revealed that the combination of melatonin and taxol influenced Snail cellular localization while promoting GSK3-β migration to the nucleus. These alterations were coupled with an increase in E-cadherin levels [121]. Likewise, studies conducted in a lung metastasis model of gastric cancer have shown that melatonin reduced the number and size of the lung metastatic nodules and reversed the induction of several EMT markers, such as vimentin, fibronectin and Snail, by interleukin 1 beta (IL-1β) [122]. Similar results were also obtained in osteosarcoma MG-63 cells, where results pointed toward a role of melatonin in suppressing the EMT via HIF-1α/Snail/MMP-9 signaling. These studies revealed that the melatonin treatment caused an increase in the ratio of E-cadherin/N-cadherin in tumor cells while suppressing Snail and MMP-9 levels [123]. Additionally, in vitro studies using human HCT15 and SW620 colorectal cancer cells showed that the melatonin treatment had a regulatory effect on E-cadherin and Snail-expression levels while exerting weak cytotoxicity [124]. Similar results were obtained in gallbladder cancer, showing an increase in epithelial markers (E-cadherin) and a decrease in mesenchymal markers (N-cadherin, vimentin and Snail), which seem to be related to the inhibition of the phosphorylation of ERK1/2 ultimately mediated by melatonin [125]. Lastly, melatonin seems to have an effect on lung cancer cell stemness by reducing the expression levels of the stem cell marker CD133 via the PLC, ERK/p38, β-catenin and Twist intracellular signaling pathways [126].

Whilst many studies have demonstrated the inhibitory effect of melatonin on Snail, its influence on other EMT-related transcription factors such as ZEB1 and ZEB2 has also been observed in cancer stem cells obtained from SKOV3 ovarian cells, which are known to show higher expression levels of EMT-related genes, such as ZEB1, ZEB2, Snail and vimentin, compared to parental SKOV3 cells. The melatonin treatment reduced the expression levels of ZEB1 and ZEB2 and some other EMT-relevant factors, thus decreasing the EMT and inhibiting the invasion properties of SKOV3-derived cancer stem cells [118].

Twist1 is also a master regulator of EMT in cancer, and it is known to play a pivotal role in the acquisition of metastatic features and angiogenesis. In the few past years, some lines of evidence have pointed to the ability of melatonin to downregulate this EMT-related TF. The first study reporting an association between Twist and melatonin was conducted in lung cancer cell lines, and the main finding was that Twist silencing increased the inhibitory action of the pineal hormone on lung cancer stemness [127]. Melatonin also suppresses lung cancer metastasis to the liver in vivo, and this action correlates with the inhibition of the EMT by downregulating Twist1 expression. The inhibitory effect of the pineal hormone on Twist1 is achieved through the binding of melatonin to its specific membrane receptor MT1. Moreover, there seems to be an association between Twist1 expression levels and tumor stage while Twist1 expression appears to be negatively correlated with MT1 expression [128].

The pineal hormone has been reported to cooperate with certain drugs commonly used in cancer treatment. As an example, melatonin seems to have a synergistic effect when administered in combination with doxorubicin, a chemotherapy agent frequently used to treat various cancer types. More specifically, combined treatment with doxorubicin and melatonin seems to inhibit the proliferation of estrogen-responsive breast cancer cells. Notably, doxorubicin alone stimulates cell migration and invasion, and this undesired effect is associated with the unexpected alteration of the transcriptional profile and miRNA expression of tumor cells. Twist 1 is known to be one of the most upregulated genes after doxorubicin treatment. Therefore, Twist1 seems to be a central player in the coordination of the cellular response to doxorubicin since several upstream modulators of TWIST1 (PTEN, Akt, IGF, p70S6K, c-Myc and TIMELESS) and downstream effectors (GLI1, Survivin, p21, Bcl-2, Bax, VEGFa, SLUG1, Per1 and Per2,) appear to be altered. Additionally, melatonin modulates the expression levels of several miRNAs that act as regulators of TWIST1, such as miR-10a, miR-10b and miR-34a, indicating that TWIST1 inhibition by the pineal hormone might be a key mechanism of overcoming drug resistance and improving the anticancer potential of doxorubicin in estrogen-responsive breast cancer [129]. Melatonin also appears to play a role in regulating the tumor microenvironment in triple-negative breast cancer cells. This effect might be explained, at least in part, by the influence of melatonin in reducing the hypoxia-inducible factor I alpha (HIF-1α) and Twist1 levels after doxorubicin treatment, leading to chemosensitivity to this anthracycline, the promotion of immunogenic cell death and the inhibition of metastasis [130]. Melatonin has also been reported to regulate the rhythmicity of Paired-related homeobox protein 1 (PRX1), a transcription factor involved in the activation of the EMT, in colorectal cancer cell lines [131]. Similarly, melatonin strongly suppresses the migration and invasion of stem cells by a potent suppression of SRY (sex determining region Y)-box (SOX9) in an in vivo model of osteosarcoma [132]. SOX9 expression also seems to be downregulated by melatonin in colorectal cancer cells [133]. The studies describing melatonin’s actions on EMT transcription factors and EMT markers, the experimental models and the melatonin concentrations are summarized in Table 1.

*Melatonin and EMT-related microRNAs*. MicroRNAs are small noncoding RNA molecules that regulate gene expression at a post-transcriptional level. Despite acting as regulators of gene expression, miRNA expression is also subjected to regulation by a wide range of intracellular signaling pathways. Many of them can promote oncogenesis while some others are considered to act as tumor-suppressor microRNAs. In terms of EMT regulation, several microRNAs are known to play a role in the modulation of metastasis through the activation of the EMT, mainly by downregulating the expression of the target genes specifically involved in the acquisition of mesenchymal features [134]. Some miRNAs have been reported to act as EMT inhibitors. Despite being considered an inhibitor of EMT-related genes, microRNA-10a can display opposite actions depending on the type of tumor subjected to its regulation. For example, microRNA-10a promotes metastasis by triggering the EMT both in vitro and in vivo in lung cancer models [57]. However, in estrogen-responsive breast cancer cell lines, microRNA-10a has been proven to act as an inhibitor of tumor progression by promoting apoptosis through the suppression of the PI3K/Akt/mTOR and p70S6K signaling pathways [135]. Moreover, melatonin has been shown to have a stimulatory effect on miRNA-10a expression in MCF-7 breast cancer cells, thus promoting apoptosis [129].

Studies conducted on a tumor spheroid invasion in vitro model of glioblastoma revealed that melatonin strongly induces the expression of miR-15b, known to act as a tumor suppressor [136]. The pineal hormone also upregulates the expression of miR-16-5p, another tumor suppressor, which in turn targets and represses Smad3 in gastric cancer cells, thus inhibiting the EMT [137]. A similar effect has been observed in LN229 cells, a human-derived glioblastoma cell line. The results obtained by using this model have shown that melatonin has an inhibitory effect on tumor growth and migration by upregulating the expression of miR-16-5p [138]. MicroRNA-34a is considered another inhibitor of the EMT, and it has been described that its protective effect against breast cancer invasion and metastasis is accomplished by the inhibition of TWIST1 [139]. In hormone-dependent breast cancer cells, melatonin treatment results in increased levels of miR-34a [129]. A similar result was obtained with breast cancer cells subjected to radiation. Ionizing radiation alone increased the expression of miR-34a, and pretreatment with melatonin potentiated this stimulatory effect [140]. Furthermore, an increase in the levels of the miR-34a/449a cluster, an miRNA involved in the regulation of Bcl-2 and Notch1, was also observed after the melatonin treatment in a xenograft model of colorectal cancer cells [141]. miR-148-3p is an example of a microRNA characterized as a tumor suppressor. The melatonin treatment appears to increase the expression levels of miR-148-3p both in vivo and in vitro, as well as decreasing IGF-R and VEGF protein levels in triple-negative breast cancer cells. Moreover, the pineal hormone inhibits survival, migration and invasion in this experimental model [142]. Further experiments in the same breast cancer cell line have revealed a role of melatonin in the upregulation of the antimetastatic miR-148v, although the depletion of this microRNA does not abolish the inhibitory effect of melatonin on migration, indicating that miR-148b is not necessary for melatonin action [143]. Additionally, the pineal hormone has also been shown to upregulate the expression levels of two more EMT suppressor microRNAs in osteosarcoma cells: miR-205 and miR-424-5p. The increase in the miR-424-5p levels mediated by melatonin results in the repression of VEGFA expression, which in turn results in the inhibition of several growth factors related to angiogenesis, suggesting a crucial role of melatonin in tumor suppression via miR-424-5p/VEGFA axis [144]. Lastly, miR-200b-3p and miR-15b, two microRNAs found to act as EMT suppressors, are upregulated by melatonin, leading to the inhibition of invasion likely by the suppression of the HIF1-α/VEGF/MMP9 signaling pathway [136].

Some microRNAs known to promote the EMT are known to be subjected to regulation by the pineal hormone. For example, oncogenic miR-21 was strongly inhibited in a thyroid cancer in vivo model. The downregulation of miR-21 resulted in a rise in PTEN levels, a proapoptotic tumor suppressor [145].

Whilst some microRNAs influence the expression of EMT master TFs, some others exert their actions downstream of the characteristic EMT factors. For example, microRNA-155, a direct target of TGF-β [61], seems to be regulated by melatonin in glioma cells. Studies carried out by using this model have suggested that the pineal hormone abolishes the cell proliferation and invasion triggered by miR-155, which is considered a stimulator of the EMT. Therefore, melatonin inhibits miR-155 expression, and this effect may be achieved via the repression of c-MYB [146]. TGF-β also upregulates miR-24, another activator of the EMT program [63]. Melatonin has been shown to reduce the expression of miR-24, reportedly known to be upregulated in both human colon carcinoma and estrogen-responsive breast cancer cell lines. The downregulation of miR-24 seems to be coupled with an inhibition of cell proliferation and invasion [147]. MicroRNA-10b is a regulator acting downstream of both TGF-β and Twist1, and it has been classified as a metastamir as it appears to play a role in promoting cell proliferation and invasion [61,148]. The treatment of estrogen-responsive breast cancer cell lines with doxorubicin strongly stimulates TWIST1 expression, consequently raising miR-10b levels. The increase in miR-10b levels seems to be a direct consequence of TWIST1 upregulation since TWIST gene silencing appears to completely abolish the induction of miR-10b expression. The strong increase in TWIST1 mediated by doxorubicin was completely reverted when cells were simultaneously treated with melatonin and doxorubicin [129].

*Melatonin and Signaling Pathways involved in the EMT*. As summarized in Section 5 of this review, the EMT is a dynamic and complex process triggered by EMT-TFs, microRNAs and many signaling pathways closely interrelated to each other [64]. TGF-β is a key regulator of embryonic development, secreted as a latent factor that, after its release and subsequent activation, binds to cell-surface receptors [65]. TGF-β has been reported to promote tumor proliferation and metastasis. This pro-oncogenic effect is achieved by the activation of several mechanisms, with the induction of the EMT being one of the key events involved in this process [66]. The TGF-β-induced EMT alters the expression of hundreds of genes, many of which are known to play a crucial role in the EMT. This effect seems to be dependent on the activation of Smad2 and Smad3 [149]. TGF-β induces the expression of the mesenchymal markers vimentin and N-cadherin and reduces E-cadherin protein levels in A549 cells, a model of pulmonary fibrosis. These effects can, however, be reversed by melatonin treatment, which also prevents the TGF-β-dependent activation of Smad2 and Smad3 by phosphorylation [150]. The pineal hormone also attenuates the EMT by preventing the release of the cytokine CCL20, thus antagonizing metastasis under hypoxic stress in glioma cells. This effect seems to be achieved by the suppression of the TGF-β/Smad-mediated increase in CCL20 [150]. Similar results have been obtained in osteosarcoma cells, where TGF-β signaling induces a switch in the E-cadherin/N-cadherin cell profile as well as an increase in vimentin, whereas melatonin appears to reverse this effect through the inhibition of Snail, MMP-9 and HIF-1α [123].

Wnt/β-catenin is another example of a signaling pathway involved in the activation of the EMT by enhancing the expression of EMT-related transcription factors. β-catenin interacts with the epithelial marker E-cadherin, creating strong cell–cell junctions. When the E-cadherin levels drop, the Wnt/β-catenin signaling becomes altered and an increase in free β-catenin in the cytoplasm will occur [73]. Melatonin has been reported to suppress the TGF-β/Smad2/3 and Wnt/β-catenin from switching the E-cadherin/N-cadherin ratio in A549 cells [151]. Wnt/β-catenin signaling has been associated with the formation of spheroids derived from colon cancer cells, which are directly associated with drug resistance and metastasis in colorectal cancer. The pineal hormone, in combination with andrographolide, has been reported to diminish cell stemness and inhibit Wnt/β-catenin signaling as well as impair several of its downstream regulatory signals, such as Cyclin D1 and l-Myc. Additionally, the levels of the VEGF transcript also seem to be downregulated in the presence of melatonin [152]. It has been recently documented that melatonin modulates stemness and migration in gastric cancer cells through the regulation of multiple signaling pathways. Additionally, melatonin appears to decrease the expression of Wnt5a, one of the several components of the canonical Wnt pathway, and β-catenin expression in a dose-dependent manner [153].

When the cell-surface receptor Notch interacts with one of its ligands, its intracellular domain is cleaved and this fragment travels to the nucleus, where it is responsible for regulating gene transcription. The Notch pathway regulates cell proliferation, differentiation and cell fate. Notch promotes the expression of important cell effectors such as NF-κB and Akt. Moreover, Notch contributes to the molecular activation of the EMT by being directly involved in Twist activation [74], increasing both Snail and Slug expression levels and repressing E-cadherin expression as well as activating β-catenin [76]. In a clinical study including women of reproductive age, it was found that compared to normal endometrium, the endometriotic eutopic endometrium shows increased expression levels of Snail, Slug and Notch. Melatonin blocks 17β-estradiol-induced migration and invasion and the EMT in normal and endometriotic epithelial cells in correlation with the decreased activity of the Notch-signaling pathway [154]. Experiments using the UMUC3 bladder cancer cell line revealed that melatonin silences the Notch/JAG2 gene and represses the PI3K/AKT/mTOR signaling, suggesting a link between both signaling pathways responsible for proliferation, invasion and metastasis in this type of cancer [155]. Furthermore, it has been reported that melatonin appears to inhibit stem-like properties in a glioblastoma cell line, and this inhibition seems to be achieved through impairing the EZH2-NOTCH1 signaling axis [156].

IL-6, a cytokine secreted by malignant cells and by nontumor cells in the surrounding tissues, activates STAT3 signaling, broadly characterized as a regulator of tumorigenesis, in mammary cancer cells and consequently weakens cell-to-cell adhesion, enhances motility and promotes the EMT [77]. IL6/STAT3 activate the EMT transcription factors Snail, Slug and Twist [78]. Studies conducted on the PCa prostate cancer cell line have shown that lipopolysaccharide stimulates invasion and promotes the EMT. One of the activated pathways in response to this bacterial compound is NF-κB/IL-6/STAT3 signal transduction. Melatonin prevents migration and invasion in this tumor cell line by blocking the EMT activation mediated by the IL-6/STAT3, AKT/GSK-3β and β-catenin pathways [157]. Tumor epithelial cells secrete cytokines that inhibit the differentiation of surrounding fibroblasts into mature adipocytes, a phenomenon known as the desmoplastic reaction [158]. In a model of desmoplastic reactions consisting of breast cancer cells cocultured with fibroblasts, melatonin reduces the levels of TNF-α, IL-6 and IL-11 mRNA expression. Decreased levels of cytokines stimulate the differentiation of fibroblasts, thus reducing the number of estrogen-producing cells proximal to malignant cells [159]. In this same model, the pineal hormone enhanced the effect of chemotherapeutic agents (docetaxel and vinorelbine) by further downregulating tumor necrosis factor α (TNFα), IL-6 and IL-11 expression [160].

The PI3K/AKT/mTOR axis stimulates the EMT process. The activation of the Akt signaling pathway results in Snail and Twist activation and E-cadherin downregulation, thus triggering the EMT [79,80]. Experiments conducted in a model of anaplastic thyroid cancer have shown that melatonin synergistically potentiated the inhibition of migration, the EMT and invasion mediated by dabrafenib through the inhibition of Akt among other mechanisms [161]. In prostate cancer cells, melatonin inhibits the phosphorylation of both Akt and its target GSK-3β, indicating that the pineal hormone inhibits the migration and invasion of PCa cells, at least in part, by suppressing the AKT/GSK-3β-mediated EMT [157]. As mentioned earlier, melatonin also represses the PI3K/AKT/mTOR signaling in bladder cancer [155]. Similar results were observed in gallbladder cancer cells, where melatonin suppresses the PI3K/Akt/mTOR signaling pathway in a time-dependent manner by inhibiting the phosphorylation of PI3K, Akt and mTOR [162]. A similar regulation of PI3K/Akt/mTOR signaling appears to occur in ovarian cancer cells, where the combined treatment with cisplatin and melatonin seems to cooperate to inhibit this signaling axis [110]. Furthermore, the combination of melatonin and alpelisib, an oral α-specific PI3K inhibitor, impairs PI3K-mediated signaling in estrogen-independent breast cancer cell lines [163].

The HH-GLI signaling pathway, triggered by HH-GLI ligands to the transmembrane receptor Patched 1 (PTCH1), promotes the EMT since its activation upregulates the expression and activates EMT transcription factors such as Snail, Slug, Twist2, ZEB1 and ZEB2 [83]. Melatonin appears to exert anticancer activity and shows synergistic effects with radiofrequency in an associated mouse lung tumor model. This effect correlates with the reduced activity of the MAPK, Wnt and Hedgehog signaling pathways [164]. The hedgehog signaling triggers an intracellular cascade activating three glioma-associated oncogenes (GLI1, GLI2 and GLI3) [82]. Experiments using hormone-dependent breast cancer cell lines revealed that melatonin partially blocks the stimulatory effect of doxorubicin on GLI1 expression, thus suggesting an antiangiogenic and anti-EMT effect mediated by the pineal hormone [129].

Melatonin can also play a role in the regulation of receptor tyrosine kinases’ signaling and ligand expression, therefore controlling EMT activation [87]. In vivo experiments conducted on a xenograft model of triple-negative breast cancer concluded that mice treated with melatonin show a significant reduction in tumor size and a lower expression of VEGFR2 compared to control animals [165]. Similarly, a melatonin treatment for human pancreatic cells (PANC-1) cocultured with human umbilical vein endothelial cells (HUVECs) prevents cell migration and diminishes both VEGF expression and secretion by malignant cells [166]. Notably, the pineal hormone diminishes VEGF expression and secretion in estrogen-responsive breast cancer cells. When cultured in the presence of HUVECs, breast tumor cells seem to stimulate endothelial cell proliferation by increasing the release of VEGF to the culture media. However, melatonin appears to revert both stimulatory effects on cell proliferation and on VEGF protein levels in the media [167]. When the endothelial cells are treated with the chemotherapeutic drugs docetaxel and vinorelbine, the expression of VEGF-A, VEGF-B, VEGF-C, VEGFR-1 and VEGFR-3 is stimulated, and the addition of melatonin counteracts this effect [168]. In the same model of cocultures of breast cancer cells and HUVECs, radiation and melatonin cooperate to reduce VEGF levels; however, radiation alone stimulates the mRNA expression of FGFR3 and IGF-1 [169]. The pineal hormone, in triple-negative breast cancer cells, upregulates the expression of miR-152-3p, therefore decreasing the protein levels of some of its target genes (IGF-1R, HIF-1α and VEGF) both in vitro and in vivo [142]. Lastly, melatonin abolishes the invasion and migration abilities of SCC-15 cells while decreasing Fibroblast Growth Factor 19 (FGF19) in oral squamous carcinoma cells (SCC-15). The addition of exogenous hFGF19 appears to revoke the effect of the indolamine on cell invasion and migration, whereas FGF19 knockdown mimics the effect of the melatonin treatment [170].

## 7. Conclusions

In cancer, the EMT is a crucial phenomenon related to the ability of tumor cells to spread to distant tissues. Cells undergoing the EMT are characterized by displaying distinctive cell features, most of them related to the loss of cell polarity, resulting in increased cell migration and invasion properties. The EMT process arises after the activation of a specific gene expression program, and this is orchestrated by several key transcription factors, mainly Snail1, Snail2 (Slug), Twist1, ZEB1 and ZEB2. Cancerous cells undergoing the EMT display a series of morphological features when acquiring the mesenchymal phenotype characterized by the increased expression of certain markers mostly associated with cell–cell adhesions such as N-cadherin, O-Cadherin, integrins, fibronectin, vimentin, FSP1 and β-catenin. The EMT program is also subjected to microRNA modulation. Some microRNAs function as repressors of certain EMT-TFs, mainly miR-10a, miR-15b, miR-29 or miR-34, whereas some other microRNAs, such as miR-21 or miR-10b, are considered EMT promoters. As a result of gene reprogramming caused by the activation of the EMT program, many intracellular signaling pathways become activated, most commonly PI3/AKT, TGF-β, IL-6/STAT3, Wnt/β-catenin, HH/Gli and RTKs.

Melatonin is a multitasking indoleamine primarily synthesized and secreted by the pineal gland. Melatonin has been extensively described as a molecule capable of preventing tumor growth at different stages, from tumor initiation and development to migration, invasion and spread. Traditionally, the role of melatonin was described by its oncostatic properties closely related to hormone-dependent tumors. However, more recent studies have highlighted the ability of this hormone to display anticancer effects in a broader range of cancers. Interestingly, a great number of biological processes involved in cancer initiation and progression are affected by the pineal hormone; thus, melatonin inhibits tumor cell growth, invasiveness, angiogenesis and metastasis while promoting apoptosis. Additionally, this indoleamine sensitizes cancer cells to conventional chemotherapeutic drugs and ionizing radiation, and this helps to overcome drug resistance. In the few clinical trials performed to date, melatonin increases the survival rates of patients while reducing the side effects caused by drugs.

The purpose of this review was to compile the current knowledge about the role of melatonin in the EMT process. As summarized in this review, there is abundant evidence demonstrating the anti-EMT actions of melatonin since the pineal hormone can inhibit the expression and activity of many of the key EMT-activating factors. Additionally, melatonin has been reported to modulate the expression of many EMT-related microRNAs while inhibiting most of the intracellular signaling pathways involved in the process, so we can conclude that melatonin has anti-EMT actions as observed in many types of cancer, with evidence arising from in vitro and in vivo studies. These findings point to the role of melatonin in the inhibition of tumor cell growth, tumor spread, migration, invasion and metastasis. The epithelial and mesenchymal markers, EMT-activating transcription factors, signaling pathways and microRNAs involved in the EMT regulated by melatonin are summarized in Figure 3.

However, to date, not many clinical trials have been carried out on the use of melatonin in cancer treatment. Since the pineal hormone has the ability to impair most of the processes related to cancer development and progression by showing protective effects against the deleterious effects of chemo- and radiotherapy and therefore improving the life quality and survival expectancies of cancer patients, we reckon that new and more extensive clinical trials combining classical or new chemotherapeutic drugs, ionizing radiation and melatonin should be designed and performed in the near future for the benefit of cancer patients.

## Figures and Tables

**Figure 1 cancers-16-00956-f001:**
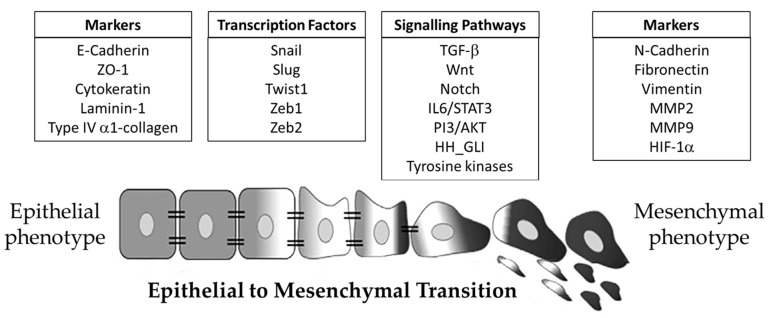
Epithelial and Mesenchymal markers. EMT is characterized by a loss of epithelial markers, such as cytokeratins and E-cadherin, and an upregulation of mesenchymal markers, such as N-cadherin, fibronectin and vimentin. EMT-activating transcription factors such as Snail, Slug, Twist1, Zeb1 and Zeb2 regulate the transition. Several signaling pathways are activated during the EMT: TGF-β, Wnt/beta-catenin. Noth. IL6/STAT3, PI3/AKT, HH-GLI and tyrosin kinases.

**Figure 2 cancers-16-00956-f002:**
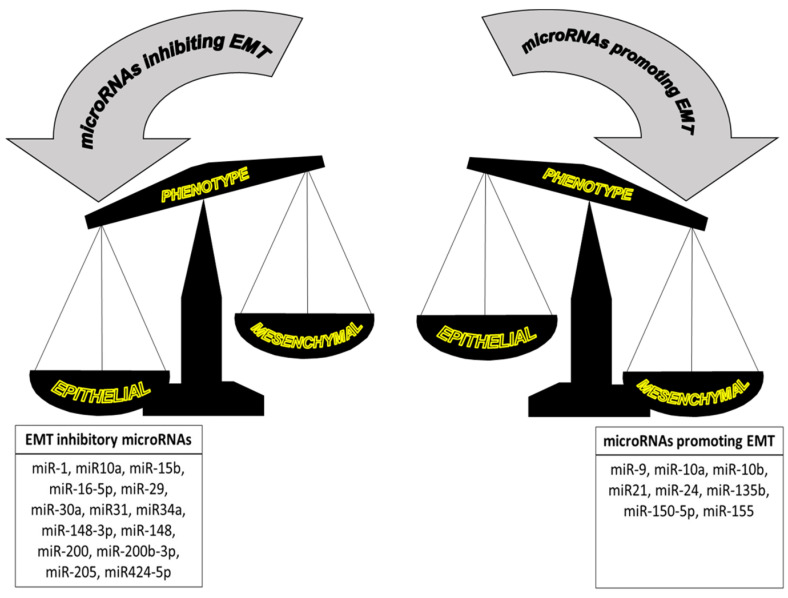
MicroRNAs involved in regulation of epithelial-to-mesenchymal transition. Many of them acting as inhibitors of the EMT and some others acting as promoters.

**Figure 3 cancers-16-00956-f003:**
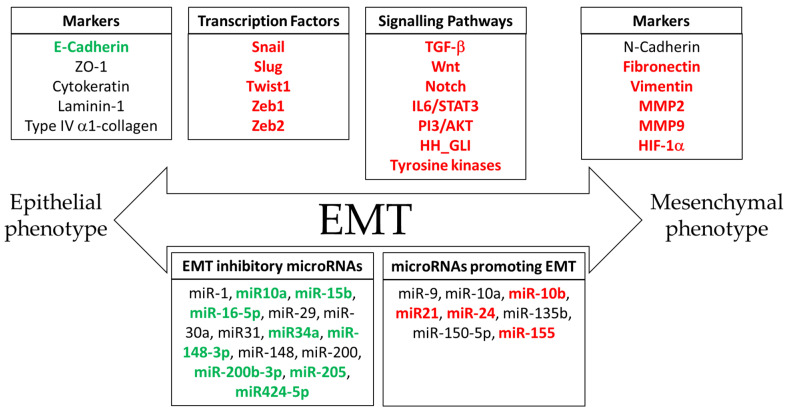
Melatonin regulates Epithelial and Mesenchymal markers, EMT-activating transcription factors, signaling pathways and microRNAs involved in regulation of epithelial-to-mesenchymal transition. In green, molecules upregulated, and in red, molecules downregulated by melatonin. The pineal hormone up-regulate the levels of the epithelial marker E-cadherin, and many of the microRNAs acting as inhibitors of EMT. By contrary, melatonin downregulates all the transcription factors promoting the EMT all the signaling pathways activated during the transition and most of the mesenchymal markers.

**Table 1 cancers-16-00956-t001:** Experimental models, melatonin concentrations and markers used in EMT studies.

Tumor	Cell Lines	In Vivo	[Mel]	Upregulated Genes	Downregulated Genes	References
Gastric cancer	AG5, MKN45		0.1 μM–2 mM	E-cadherin	MMP2, MMP9, NF-κB, Snail	[115]
Gastric cancer	MGC80-3, SGC790		0.1–1.5 mM	E-cadherin	Snail, Slug, Wnt/β-catenin, NF-κB	[116]
Gastric cancer	MGC80-3	In vivo lung metastasis	100 mg/Kg/day	E-cadherin	MMP2, MMP9, NF-κB, Snail, Slug, Fibronectin	[122]
Ovarian cancer	SK-OV-3		3.4 mM		ZEB1, ZEB2, Snail, Vimentin	[118]
Ovarian cancer	SK-OV-3	Tumor-bearing mouse model	200 μg/100 g/day		Akt, β-catenin, Slug	[119]
Bladder cancer	UC3		1 μM	E-cadherin	N-cadherin, Fibronectin, Snail, Slug	[117]
Oral cancer	SCC25, SCC9, Tca8113, Cal27, FaDu		1 mM		Akt, Snail, Vimentin	[120]
Breast cancer	MCF-7, MDA-MB-231		1 nM	E-cadherin	Snail	[121]
Breast cancer	MCF-7		1 nM		Twist1, Slug, Akt	[129]
Osteosarcoma	MG-63		200 nM	E-cadherin	N-cadherin, Snail, MMP-9	[123]
Colorrectal cancer	HCT-15 SW620		1–2 mM	E-cadherin	Snail	[124]
Gallbladder cancer	GBC-SD		0.1–2 mM	E-cadherin	N-cadherin, Snail, Vimentin	[125]
Non-small cell lung cancer	H1299		0.1 mM		Snail, Twist1	[126]
Lung cancer	A549, CL1-5		1 mM		Twist1	[127]
Lung cancer	CL1-5		1 mM		Slug, Twist1, β-catenin	[128]

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
