# Peer review of "Melatonin and Its Role in the Epithelial-to-Mesenchymal Transition (EMT) in Cancer"

_cancers, 2024, doi:10.3390/cancers16050956_

Round 1

Reviewer 1 Report

Comments and Suggestions for Authors

This review manuscript compiled data on the ability of melatonin to inhibit epithelial-to-mesenchymal (EMT)-related biomarkers, transcription factors, and EMT-related micro-RNAs. Authors summarized the state-of-the-art concerning the multiple pathways by which melatonin may control the EMT in different tumor types.

 Although important and well-constructed, this review could be benefited in some aspects

1)     English is good but could be improved in particular sentences. Also, correct some grammar errors and many “typos” over the text.

2)     The topic “epithelial-to-mesenchymal transition” should be focused on “type 3 classification” avoiding unnecessary descriptions of other types.

3)     On page 16, the last sentence on VEGF signaling has no mechanistic involvement with EMT process. Authors may clarify them or remove the description.

4)     Authors also could generate a table with references to experimental models (cell line or animal), melatonin concentrations/dosages, and summarized markers that were used in the studies of EMT.

5)     The first paragraph of conclusion is too general and focused on antitumor actions of melatonin and not EMT.

Minor comments

- The sentence “Besides its widely recognized effects on hormone-dependent tumors, melatonin..........and hematopoietic cancers, among others” is similar to one previously used in the first paragraph of the text with no citation. Avoid repeated information when possible.

- Utilize proper gene abbreviations for humans (see gene cards). For humans, gene is abbreviated using uppercase letters and in italic.

- Page 4, line 175. The phrase “subtypes based on….” is lacking completion.

- Page 13, line 616, Capitalize “MicroRNA-10a” instead of “microRNA-10a”

- The sentence “microRNA-10a 616 has been described to promote metastasis by stimulating EMT both in vitro and in vivo 617 [57], although in estrogen-responsive breast cancer cells lines, it has been proven to act as 618 an inhibitor or tumor progression by promoting apoptosis through suppression of the 619 PI3K/Akt/mTOR signaling pathway and p70S6K [138]; melatonin has a stimulatory effect 620 on its expression [132]…” is confusing and should be improved for readability.

- Correct the terms “MicroRNA miR-155,...” and “MicroRNA miR-10b”.

Comments on the Quality of English Language

Well-written but it contains grammar errors and some typos

Author Response

Thank you very much indeed for your kind and useful evaluation of our manuscript entitled ": Melatonin as its role in the Epithelial-to-Mesenchymal Transition (EMT) in Cancer”. We are sending you the revised manuscript.

Please find below the response to the reviewers’ comments in the Word file

Reviewer 2 Report

Comments and Suggestions for Authors

Comments

1.       The abstract is too long, please make it briefly.

2.       Figure (1) and (2) is very simple, needs to make it attractive.

3.       The legends for both figures, needs to be more elaborative.

4.       The conclusion is very lengthy; I would suggest to make it precise.

5.       Since, its long review, I would suggest to add more figures to make the section, easily understandable.

6.       Please add tables also, especially for microRNAs.

Comments on the Quality of English Language

Extensive english is required.   

Author Response

Thank you very much indeed for your kind and useful evaluation of our manuscript entitled ": Melatonin as its role in the Epithelial-to-Mesenchymal Transition (EMT) in Cancer”. We are sending you the revised manuscript.

Please find below the response to the reviewers’ comments in the PDF file

Round 2

Reviewer 1 Report

Comments and Suggestions for Authors

No additional comments

Reviewer 2 Report

Comments and Suggestions for Authors

Authors addressed my concerns.